# Dietary Isothiocyanates: Novel Insights into the Potential for Cancer Prevention and Therapy

**DOI:** 10.3390/ijms24031962

**Published:** 2023-01-19

**Authors:** Guanqiong Na, Canxia He, Shunxi Zhang, Sicong Tian, Yongping Bao, Yujuan Shan

**Affiliations:** 1Department of Nutrition, School of Public Health and Management, Wenzhou Medical University, Wenzhou 325035, China; 2Southern Zhejiang Institute of Radiation Medicine and Nuclear Technology, Wenzhou Medical University, Wenzhou 325035, China; 3Institute of Preventative Medicine, School of Medicine, Ningbo University, Ningbo 315211, China; 4Norwich Medical School, University of East Anglia, Norwich NR4 7UQ, UK

**Keywords:** isothiocyanates, sulforaphane, cancer, angiogenesis, microbiota, stem cells, tumor microenvironment, rearrangement of energy metabolism

## Abstract

Diet plays an important role in health. A high intake of plant chemicals such as glucosinolates/isothiocyanates can promote optimal health and decrease the risk of cancer. Recent research has discovered more novel mechanisms of action for the effects of isothiocyanates including the modulation of tumor microenvironment, the inhibition of the self-renewal of stem cells, the rearrangement of multiple pathways of energy metabolism, the modulation of microbiota, and protection against *Helicobacter pylori*. However, the hormetic/biphasic effects of isothiocyanates may make the recommendations complicated. Isothiocyanates possess potent anti-cancer activities based on up-to-date evidence from in vitro and in vivo studies. The nature of hormesis suggests that the benefits or risks of isothiocyanates largely depend on the dose and endpoint of interest. Isothiocyanates are a promising class of cancer-preventative phytochemicals, but researchers should be aware of the potential adverse (and hormetic) effects. In the authors’ opinion, dietary isothiocyanates are better used as adjunctive treatments in combination with known anti-cancer drugs. The application of nano-formulations and the delivery of isothiocyanates are also discussed in this review.

## 1. Introduction

Dietary isothiocyanates (ITCs) are the breakdown products of glucosinolates, which occur almost exclusively in cruciferous vegetables. Epidemiological studies have shown an inverse association between a higher intake of these vegetables and a lower incidence of many types of cancer [1]. Many studies have shown ITCs to be potent inhibitors of tumorigenesis both in vitro and in vivo. Early investigations have demonstrated that ITCs can inhibit phase I carcinogen-activating enzymes, induce phase II detoxification enzymes, induce cell cycle arrest, induce apoptosis (and autophagy), inhibit angiogenesis, invasion and metastasis [2,3,4,5].

It must be emphasized that ITCs are potent inducers of the Nrf2-Keap1-ARE pathway and act as powerful regulatory agents regarding the cellular redox status. The induction of Nrf2 plays a critical role in the protection against free-radical or mutagen-mediated oxidative stress and cell damage [6,7,8,9]. However, Nrf2 also plays a dual role in carcinogenesis, which is associated with the hormetic nature of ITCs [10].

In the past decade, quite a few novel mechanisms for ITCs in protection against carcinogenesis have been proposed (Figure 1), which include the modulation of the tumor microenvironment [11], the inhibition of self-renewal of stem cells [12], the rearrangement of energy metabolism [13,14] and the regulation of microbial homeostasis [15]. On the basis of these findings, this review evaluates the current state of knowledge on the doses of ITCs/sulforaphane (SFN) and broccoli that have been used in cell, animal, pre-clinical and clinical studies. We aim to fill in the gaps in the understanding of the pleiotropic effects of ITCs and, especially, their hormetic bioactivities-both the benefits and potential risks (if there are any), and under what conditions these ITCs should potentially be avoided. Finally, the nano-delivery of ITCs and the great potential for these nano-compounds to act as adjuvant drugs will be discussed. More studies in the future are required to elucidate the underlying mechanisms of action and to develop and validate biomarkers for both their pharmacokinetics and efficacy in humans.

## 2. Microbiota and Anti-Cancer Effects of ITCs

### Microbiota: A Mediator Transforming Glucosinolate Precursors to the Active Isothiocyanates

ITCs mainly exist in cruciferous vegetables in the form of glucosinolate (GLS) precursors. Myrosinase (thioglucoside glucohydrolase EC3.2.1.147, formerly EC 3.2.3.1) [16], normally found in plant cells but not in the human body, is the only known endogenous enzyme that degrades the GLS precursors into their corresponding active forms-ITCs [17], mainly including sulforaphane (SFN), allyl isothiocyanate (AITC), benzyl isothiocyanate (BITC) and phenethyl isothiocyanate (PEITC). The dietary source and chemical structures of ITCs are summarized in Figure 2. However, endogenous myrosinase can be largely inactivated during cooking, which hampers the degradation of GLSs to ITCs and, thus, their beneficial effects. The gut microflora may provide an outstanding substitute for endogenous myrosinase following the consumption of cooked cruciferous vegetables. Therefore, for people who often consume cooked cruciferous vegetables, the acquisition of ITCs in their bodies mainly depends on the intestinal microecology.

The microflora of the gastrointestinal tract is thought to be primarily responsible for ITCs production in the human body. ITCs were shown to appear in the mesenteric plasma 120 min following the injection of glucoraphanin (GRP) into the cecum (150 μmol/L), further confirming the capabilities of the gut microflora to hydrolyze GLSs [18]. *Peptostreptococcus* spp. and *Bifidobacterium* spp. have been proven to be able to metabolize GLSs [19,20]. Two beneficial bacteria, *Lactobacillus plantarum* KW30 and *Lactococcuslactissubsp. lactis* KF147, were shown to be capable of transforming 30–33% of GRP and/or glucoerucin into SFN nitrile, erucin nitrile and some unknown metabolites [21]. Another lactic acid bacterial strain, *Lactobacillusagilis* R16, possessed an outstanding ability to degrade sinigrin into allyl isothiocyanate (AITC) [19]. Some other microorganisms, such as *Bacillus cereus* and *Aspergillus niger*, have also presented myrosinase activity [19,20,21,22]. A strain of lactic acid bacteria derived from the traditional fermented foods of Xinjiang Province, China, that shows excellent myrosinase capability has been isolated by our group (unpublished).

## 3. Activity of ITCs on *Helicobacter pylori* Strains

*Helicobacter pylori* infections affect more than 50% of the world’s human population and are associated with gastritis, peptic ulcer disease and gastric cancer [23]. Gastric *H. pylori* infections express high urease activity, which promotes inflammation. ITCs have potent antibacterial effects against gastric *H. pylori*, and the MIC90 values of these ITCs range between 4 and 32 µg/mL [15]. SFN is a potent bacteriostatic agent that is effective against three reference strains and 45 clinical isolates of *H. pylori* (the MICs for 90% of the strains are <4 µg/mL). Furthermore, SFN prevented the formation of benzo[a]pyrene-induced stomach tumors in ICR mice [24]. In human gastric xenografts that were implanted in nude mice, SFN significantly inhibited the proliferation of *H. pylori*, with an observed eradication rate of 73%, which depended on the Nrf2-dependent induction of phase II detoxication and antioxidant enzymes. Moreover, broccoli sprouts (0.125 mg/mL, *w*/*v*) reduced *H. pylori*-associated inflammation by inhibiting IL-8 release in response to TNFα exposure [25].

Some population intervention studies have also suggested that ITCs can reduce *Helicobacter pylori* infections. Among nine patients who consumed broccoli sprout preparations (14, 28 or 56 g fresh weight) twice daily for 7 days, the eradication of *H. pylori* infection was observed in three [26]. Forty-eight *H. pylori*–infected patients were randomly assigned to the feeding of broccoli sprouts (70 g/d; containing 420 μmol of SFN precursor) for 8 weeks or to the consumption of an equal weight of alfalfa sprouts (not containing SFN) as a placebo. The intervention with broccoli sprouts (BS) decreased the levels of urease measured and serum pepsinogens I and II [27]. In another trial involving 25 *H. pylori*–positive volunteers, eight subjects had *Helicobacter pylori* stool antigen (HpSA) values below the cutoff (0.100) at the end of the 8 week BS treatment (70 g/day) period. In six of these subjects, the HpSA values became positive again at 8 weeks after the cessation of BS consumption, indicating that BS treatment reduced *H. pylori* colonization but did not result in complete eradication [28].

SFN protected against small intestinal injury induced by nonsteroidal anti-inflammatory drugs, through inhibiting the invasion of anaerobic bacteria into the mucosa [29]. Our results showed that SFN prevented *N*-butyl-*N*-(4-hydroxybutyl)-nitrosamine (BBN, 0.1%)-induced bladder cancer through normalizing the abnormal intestinal flora in C57BL mice [30]. The related mechanisms included restoring the balance of the gut microbiota, inducing the production of fecal butyric acid and the expression of Gpr41 and Glp2 downstream, and ameliorating the mucosal damage.

## 4. Rearrangement of Energy Metabolism Phenotype by Sulforaphane

Malignant cells depend on glycolysis to provide energy, which produces lactate and forms an acidic tumor microenvironment, which finally favors tumor metastasis and chemoresistance. SFN prevents the propagation of bladder cancer cells by the inhibition of glycolysis, mediated by the downregulation of hypoxia inducible factor 1 and its nuclear translocation [13]. Furthermore, SFN prevented the increase in glycolysis, hexokinase and pyruvate kinase activity, and reduced the HIF-1α stabilization induced by an androgen and Tip60 in LNCaP cells [31]. SFN weakens the glycolytic flux by suppressing multiple metabolic enzymes, including hexokinase 2 (HK2) and pyruvate dehydrogenase (PDH). Moreover, SFN significantly downregulated glycolysis and mitochondrial OXPHOS via blocking the AKT1/HK2 axis and PDH expression [32]. SFN significantly downregulated the expression of hexokinase II (HKII), pyruvate kinase M2 and/or lactate dehydrogenase A (LDHA) in vitro and in vivo in neoplastic lesions in the prostates of transgenic adenocarcinoma of mouse prostate (TRAMP) and Hi-Myc mice, and significantly suppressed glycolysis in the prostates of Hi-Myc mice [33].

Another metabolic hallmark of malignant cells is the dysregulation of lipid metabolism, such as the enhancement of de novo fatty acid synthesis, lipid absorption and steatolysis, which provide energy for the rapidly proliferating malignant cells. The treatment of prostate cancer cells with SFN (5 and 10 μM) resulted in the downregulation of acetyl-CoA carboxylase 1 (ACC1) and fatty acid synthase (FASN), inhibiting fatty acid synthesis. Similarly, SFN administration to TRAMP mice resulted in a significant decrease in the plasma and/or prostate adenocarcinoma levels of total free fatty acids, total phospholipids, acetyl-CoA and ATP. Additionally, SFN prevented fatty acid absorption by decreasing carnitine palmitoyl-transferase 1A (CPT1A), followed by the β-oxidation of fatty acids [14]. SFN downregulated FASN, acetyl CoA carboxylase (ACACA), and ATP citrate lyase (ACLY) via activating proteasomes and downregulating the transcriptional factor SREBP1. SFN also decreased the amount of intracellular fatty acid and inhibited microtubule-mediated mitophagy, leading to apoptosis in human non-small-cell lung cancer (NSCLC) cells [34].

## 5. Inhibition of Cancer Stem Cells by Sulforaphane

Cancer stem cells (CSCs) are a small group of cancer cells present in malignant tumors that are characterized by high drug resistance, potent metastatic and invasive ability, and infinite self-renewal. Mounting evidence confirms that CSCs are the root cause of the initiation and relapse of malignant tumors [35]. SFN inhibited CSC self-renewal via the blockade of Wnt/β-catenin, Hedgehog and Notch signaling. SFN reduced aldehyde dehydrogenase (ALDH)-positive CSCs by 65–80% and mammosphere formation, suggesting that it is effective in eliminating breast CSCs. Such inhibitory effects were due to the attenuation of β-catenin-mediated transcription [12]. SFN significantly inhibited spheroids derived from human pancreatic CSCs by inhibiting sonic hedgehog (Shh) pathway components and Gli transcriptional activity (PDGFRα and Cyclin D1). Thus, the preventative effects of SFN on pancreatic cancer may result from the inhibition of the Shh pathway [36]. In addition, the stemness of pancreatic CSCs was repressed by SFN in vitro and in vivo, which is characterized by the inhibition of tumor-sphere formation and of the expression of pluripotency maintaining factors (Oct4 and Nanog) [37]. Recent studies have also found that YAP1, the Hippo signaling transcription adaptor protein, and ∆Np63α, a key epidermal stem cell survival protein, form a complex to drive epidermal cancer stem cell survival. However, sulforaphane suppresses epidermal squamous carcinoma cell formation and this is associated with reduced levels of YAP1 and ∆Np63α [38].

The modulation of microRNAs contributes to the anti-CSC effect of SFN [39]. SFN exhibited inhibitory effects on lung CSCs through suppressing miR-19 and the Wnt/β-catenin pathway [40]. SFN increased exosomal miR-140 expression and decreased both miR-21 and miR-29, which led to reduced ALDH1 levels and decreased mammosphere formation [41]. In addition, SFN restored miR-140 expression, inhibited the tumorigenicity of MCF10DCIS (a model cell line of poorly differentiated basal-like ductal carcinoma in situ) stem-like cells in immune-deficient nude female mice, and decreased the percentage of CSCs (CD44^high^/CD24^low^) in MDA-MB-231 cells [42]. SFN downregulated Bmi1 via increasing miR-200c expression, thereby blocking the expression of CSC markers and, subsequently, the migration and invasion of oral squamous CSCs [43].

## 6. Remodeling of the Tumor Microenvironment by Sulforaphane

In the tumor microenvironment, cancer cells bound to the T cell co-inhibitory CD28 family receptors including programmed cell death receptor 1 (PD-1) and cytotoxic T-lymphocyte-associated protein 4 (CTLA-4), which decreased immune cell activity and allowed cancer cells to escape from immune surveillance [44]. The activities of PD-1 and its ligand PD-L1 are responsible for reducing activation, proliferation, and cytokine secretion of T cells in TME, resulting in decreased anti-tumor immune responses [45,46]. SFN inhibited the transformation of normal monocytes to myeloid-derived suppressor cells (MDSCs) by glioma-conditioned media in vitro, and promoted the proliferation of T cells. SFN also suppressed PD-L1 expression in a dose-dependent manner [47]. By inducing miR-194-5p (targeting the B7-H1 gene) and inhibiting miR-155-5p expression, SFN attenuates the phosphorylation of STAT3 in dendritic cells. The inhibition of JAK/STAT3 leads to the downregulation of B7-H1 expression, thereby promoting the activation of T cells [48].

Epithelial–mesenchymal transition (EMT) is a highly conserved program in which epithelial cells alter their morphology and gain the properties of mesenchymal cells. This process is critical for initiation, invasion and metastasis in a variety of cancers [49]. The inhibition of anchorage-independent growth, invasion, and migration of the endometrial cancer cell lines by SFN was associated with the increased E-cadherin and the decreased N-cadherin and vimentin expression [50]. Other results showed that SFN prevented EMT via inducing miR-200c [51], and dose-dependently induced the EMT hallmark E-cadherin, and downregulated vimentin and the transcriptional repressor ZEB1, which depended on the regulation of miR-200c [52]. In addition, SFN reversed mesenchymal-like changes induced by TGF-β1 and restored cells to their epithelial-like morphology. SFN increased the expression of E-cadherin, while decreased the expression of the mesenchymal markers including N-cadherin, vimentin, and α-SMA in A549 cells. SFN inhibited TGF-β1-induced mRNA expression of the EMT-related transcription factors, Slug, Snail, and Twist [53].

## 7. The Hormetic Effects of ITCs on Cancer

The term hormesis is commonly used by toxicologists to describe a biphasic dose/concentration response characterized by a low-dose stimulation and a high-dose inhibition. A hormesis database covering approximately 9000 dose relationships for nearly 2000 different agents from approximately 245 different chemical classes was developed by Calabrese and Blain [54,55]. Recently, a substantial volume of scientific literature has indicated that many phytochemicals such as resveratrol, curcumin, ginseng, isothiocyanates, *ginkgo biloba*, genistein, daidzein and ferulic acids [56,57] exhibit biphasic dose/concentration responses.

ITCs have been shown to exhibit hormetic effects on cell growth, migration and even tumor growth [58,59]. Low doses of SFN (1–5 µM) promoted cell growth by 20–43% compared to the control, whereas high doses of SFN (10–40 µM) inhibited cell growth in numerous tumor cell models including bladder cancer T24, hepatoma HepG2, and colon cancer Caco-2 cells [58]. The results from Nrf2 knockdown or thioredoxin reductase-1 (TrxR-1) experiments suggest that Nrf2- or TrxR-1-induced cell death might be related to the biphasic effects of SFN on cancer cell growth [60]. The hormetic effects of ITCs on tumor growth have also been observed in in vivo models. In an BBN-induced bladder cancer mouse model, low doses of SFN (2.5 and 5 mg kg^−1^ BW) led to more mice dying and an increase in bladder weight due to tumor occurrence in bladder tissue [30].

After treatment for 24 h, 2.5 µM AITC decreased the DNA damage in HepG2 cells from 21.57 to 12.30%, while 10 and 20 µM AITC increased it to 36.12% and 47.48%. Further studies revealed that the DNA repair protein Ku70 was involved in this biphasic effect of AITC on DNA integrity [59]. The Nrf2 signaling pathway plays an essential role in the hormetic effects of ITCs on cell genomic instability and cell migration. The promoting effects of 2.5 µM AITC on cell migration and DNA damage were abolished after knockdown of the Nrf2 gene or the inhibition of glutathione (GSH) synthesis in HepG2 cells [61]. Nrf2 siRNA and the GSH-depleting agent L-buthionine-sulfoximine also abolished the effect of SFN-promoted T24 cell growth [unpublished]. SFN at 0.5–5 µM caused a 39–140% increase in GSH content; however, high dose SFN (10 µM) decreased the GSH level to 50% of the control value [62].

Cell migration involves a multistep process such as the extravasation of plasma proteins, the degradation of the extracellular matrix, endothelial cell migration and proliferation, and angiogenesis [63]. SFN at 2.5 and 3.75 µM increased T24 cell migration by 28–33% compared to the controls. Results published by our group suggested that high doses of SFN (>5 µM) significantly inhibited the ability of HUVECs to form a capillary-like tubular structure in 3D co-culture with pericytes, and blocked microvessel sprouting from mouse aortic rings ex vivo-all of which are critical steps in angiogenesis-by inhibiting the STAT3/HIF-1α/VEGF signaling pathway [59]. At lower concentrations (1.25–5 μM), SFN increased hypoxia-induced HUVEC migration and tube formation, and alleviated the hypoxia-induced inhibition of proliferation, but higher doses (≥10 μM) showed an opposite effect through an Nrf2-dependent mechanism [64]. The activation of autophagy by SFN may be responsible for the hormetic effect of SFN on cell migration. An autophagy inhibitor, 3-methyladeine, reduced the cell migration induced by 2.5 µM SFN from 128 to 26% [58]. A low dose of AITC (2.5 µM) significantly increased HepG2 cell migration (130% compared to the control) after 48 h, while a high dose of AITC (20 µM) inhibited it by nearly 30% [59].

When using chick yolk sac membrane and chorioallantoic membrane models, low doses of SFN (2.5–10 μM) alone increased angiogenesis and high concentrations of SFN (20–40 μM) inhibited angiogenesis [65]. Besides SFN, other ITCs also showed biphasic effects on angiogenesis. For example, AITC treatment (>5 µM) sharply decreased the total length of formed tube, while lower doses of AITC (1.25 and 2.5 μM) significantly promoted the formation of capillary tubular structures [61].

It should be noted that the hormetic-zone concentrations of ITCs (~5 µM) in cell culture studies could readily be achieved in the human plasma following the consumption of a meal rich in cruciferous vegetables or supplements [58,66]. More importantly, our findings further suggest that the hormetic effects of ITCs display little cellular selectivity [58]. Thus, it is crucial to optimize the anti-cancer effects and minimize the potential risks of ITCs in cancer chemoprevention or treatment. The mechanistic profiles of ITCs regarding their hormetic dose responses indicate that the activation of the Nrf2/ARE pathway is probably a central, integrative, and underlying mechanism. Nrf2 conducts crosstalk with other redox-activated transcription factors to further amplify the significance of Nrf2 activation in mediating a hormetic response [10]. All the data above suggest that the role of Nrf2 in cancer development is controversial, and Nrf2 activators such as ITCs may show both benefits and drawbacks regarding cancer development. Therefore, a deeper understanding of the hormesis-mediated activation of Nrf2 by ITCs and the mechanisms should enable the exploitation and development of hormetic effects for application in protecting against cancer and maintaining health.

## 8. Nanotechnology-Based ITC-Delivery Systems

www.clinicaltrial.gov currently lists more than 50 clinical trials examining the pharmacokinetics, pharmacodynamics and disease-mitigating effects of ITCs for different health issues including non-alcoholic fatty liver disease (NAFLD), metabolic syndrome (MS), inflammation, radiation dermatitis, osteoarthritis, autism, schizophrenia, depressive disorder, Parkinson’s disease, Alzheimer’s disease, chronic obstructive pulmonary disease (COPD), cardiac and vascular dysfunction, asthma, chronic kidney disease, cystic fibrosis, *Helicobacter pylori* infections and various cancers (including breast, prostate, colon, lung, pancreatic and bladder cancers) at https: www.clinicaltrial.gov, accessed on 30 November 2022. Due to the complex pathological environments of tumors, the clinical utilization of ITCs to combat cancer faces numerous challenges, including their instability, their poor aqueous solubility, their low bioavailability, the difficulty in overcoming multiple biological barriers and insufficient tumor penetration. Accordingly, much effort has been dedicated to ITCs-nano-delivery approaches to address these essential problems [67,68,69,70,71].

Several prior publications employed conventional nanoparticles as carriers for transporting ITCs and/or other compounds with synergistic effects, such as liposomes [67,72,73], micelles [74,75], polymeric nano-particles [68,76,77], solid lipid nanoparticles [69] and a few inorganic nanomaterials [70,71,78]. These ITCs nano-delivery systems are summarized in Figure 3. A recent study applied tri-block copolymers to establish an efficient SFN-delivery nano-system, which enabled a remarkably sustained release of SFN in vivo. The nanostructures possess passively tumor-targeting and recycling properties due to the enhanced penetration and retention (EPR) of tumor blood vessels [75]. ITCs have also attracted extensive attention as synergistic agents in combination with other anti-cancer drugs. SFN and cisplatin (CDDP) were encapsulated in liposomes (approximately 130 nm in size) to sensitize non-small-cell lung cancer (NSCLC) [79]. The resulting doublet formulation showed strongly enhanced toxicity towards NSCLC cells. Furthermore, a polymeric nanoparticle co-delivering SFN and a water-soluble CDDP derivative was constructed to achieve efficient GSH depletion and the effective accumulation of CDDP in tumor cells [80]. The results showed that the nano-drug (SFN-CDDP-NPs) exhibited significantly higher tumor accumulation and greater anti-tumor activity compared to CDDP alone. To increase the potency, a multifunctional nanostructure of a carbon dot with a thiourea skeleton carrying SFN was built to target epidermal growth factor receptor (EGFR)-overexpressing tumor cells for imaging and inhibition [81]. In other research, gold-coated nanoparticles modified with both thiolated polyethylene glycol–folic acid and FITC were designed and fabricated. Folic acid can play a role in targeting cancer tissue, while iron oxide has the potential to facilitate magnetic resonance imaging in vitro [82].

Although the above mentioned SFN-based nano-delivery strategies enhanced the effectiveness of ITCs, simple modification countermeasures made the compounds subject to biological barriers, reticuloendothelial clearance, and finite penetration into tissues. It is difficult to realize absolutely targeted delivery or refined therapy. It is therefore essential to develop novel nanomaterials with particular properties (especially natural nanocarriers with low immunogenicity, outstanding biodegradation and biocompatibility, and distinguished pharmacokinetics, such as peptides, nucleic acids, cell membranes and vesicles) and apply multifunctional modification designs sagaciously. Nanorobots loaded with anti-tumor molecules have recently garnered great interest and research focus. An ideal nanorobot delivery system should contain a medicine-loaded structural framework, tumor-targeted shell and responsive module. These elements simultaneously enable the overcoming of biological barriers, the specific targeting of and enrichment in tumor tissues, resistance to the impact of blood flow and increased tumor penetration.

Diverse types of cancers, different individuals and different regions within tumors have different physiological characteristics. Precision therapy (or personalize medicine) is a megatrend in drug-delivery strategies. According to the specific requirements, the reasonable design of a nano-delivery scheme loaded with ITCs is a focus of exploration. Of course, evaluating the preclinical safety and effectiveness of nanodrugs is also indispensable.

## 9. Conclusions

A recent umbrella review suggests an inverse association between cruciferous vegetable intake and the risk of gastric cancer, lung cancer, endometrial cancer, and all-cause mortality [83]. Here, we summarize the main studies on antitumor effects of ITCs (Table 1) and present more mechanistic evidence that dietary ITCs target tumorigenic processes, with effects on the tumor microenvironment, stem cells, mitochondrial function, energy metabolism and the microbiota. In our opinion, ITCs are a promising class of cancer-preventative phytochemicals, but researchers should be aware of the potential adverse and/or hormetic effects of isothiocyanates. The benefits or risks are largely dependent on the dosage and endpoint of interest; for example, the promoting effects of low doses could be beneficial for cardiovascular health and the formation of new blood vessels.

There are more ongoing clinical trials listed at www.clinicaltrial.gov. ITC-based nano-delivery approaches were tested to overcome the essential problems of ITCs in clinical utilization. Based on the current views on the application of ITCs in cancer therapy, we believe that ITCs are better used as adjunctive treatments in conjunction with known anti-cancer drugs, conferring several advantages: (1) reducing the side effects of anti-cancer drugs; (2) improving the drug efficacy; and (3) overcoming chemoresistance. Therefore, the application of nano-formulations and the specific targeted delivery of ITCs may be other avenues to explore over the next decade.

## Figures and Tables

**Figure 1 ijms-24-01962-f001:**
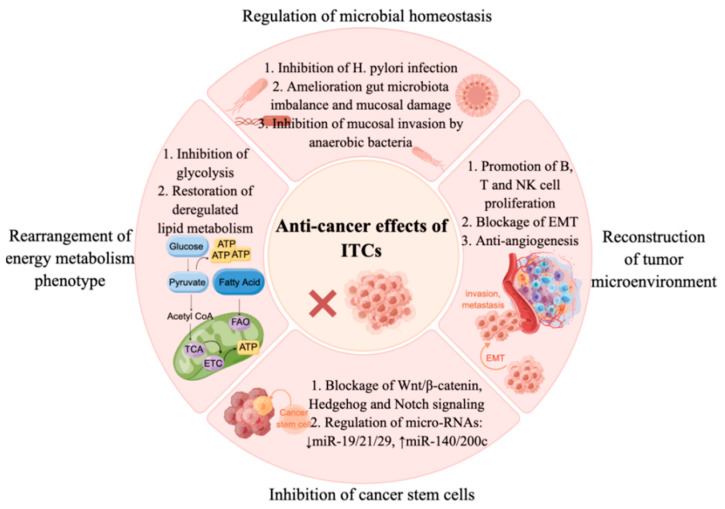
Multiple mechanisms of isothiocyanates in cancer prevention and therapy. ITCs inhibit cancers through regulating microbial homeostasis, such as inhibiting *H. pylori* infection, ameliorating gut microbiota imbalances and mucosal damage, and inhibiting mucosal invasion by anaerobic bacteria. ITCs inhibit cancer cell proliferation via suppressing glycolysis, restoring deregulated lipid metabolism, and inhibiting CSCs via regulating self-renewal signaling (Wnt/β-catenin, Hedgehog and Notch signaling) and miRNA pathways. Furthermore, ITCs exert anti-cancer effects via remodeling the tumor microenvironment, such as promoting B, T and NK cells’ proliferation, blocking EMT and inhibiting angiogenesis. ITCs, isothiocyanates; *H. pylori*, *Helicobacter pylori*; miR, microRNAs; EMT, epithelial-to-mesenchymal transition; TCA, tricarboxylic acid cycle; ETC, electron transport chain; FAO, fatty acid oxidation; NK cell, natural killer cell.

**Figure 2 ijms-24-01962-f002:**
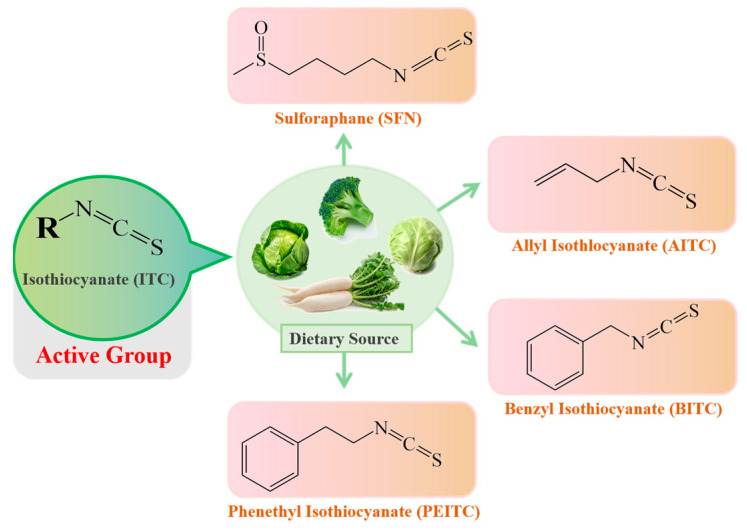
The dietary source and basic structure of ITCs.

**Figure 3 ijms-24-01962-f003:**
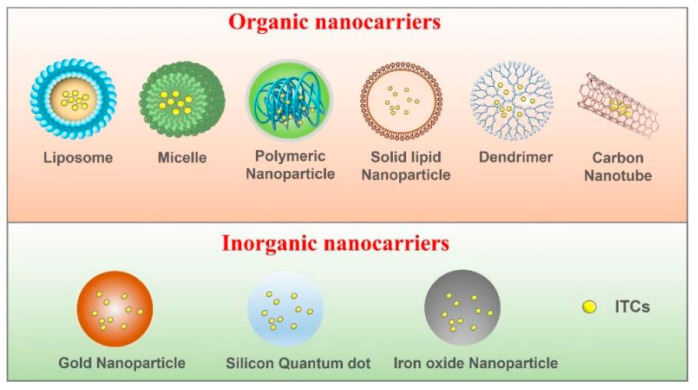
Schematic diagram of major ITCs nano-delivery systems for cancer treatment.

**Table 1 ijms-24-01962-t001:** Summary of main research on antitumor effect of ITCs.

Isothiocyanate	Disorders	Research Type	Information/Results	Reference
SFN	Breast cancer	In vitro	ALDH positive CSCs was reduced by 65–80% by SFN and breast CSCs can be effectively eliminated.	[12]
SFN	Liver cancer	In vitro	SFN is a potent inducer of apoptosis in hepatocellular carcinoma cells via PFKFB4 inhibition pathways.	[13]
broccoli sprouts/SFN	*Helicobacter acceleratum*/ gastric cancer	In vitro	*H. pylori*-associated inflammation was inhibited by reducing IL-8 releasing.	[25]
broccoli sprouts/SFN	*Helicobacter pylori* infection	Clinical trial	*Helicobacter pylori* infections were reduced significantly.	[26]
broccoli sprouts/SFN	*Helicobacter pylori* infection	Clinical trial	Urease and serum proproteinase I and II levels were significantly reduced.	[27]
broccoli sprouts/SFN	*Helicobacter pylori* infection	Clinical trial	*Helicobacter pylori* stool antigen (HpSA) values became negative after 8-week Treatment.	[28]
SFN	Bladder cancer	C57BL mice	Bladder carcinogenesis was effectively prevented by SFN via balancing of intestinal flora.	[30]
SFN	Prostate cancer	In vitro and in vivo	The expressions of hexokinase II, pyruvate kinase M2 and/or lactate dehydrogenase A were significantly down-regulated and glycolysis was inhibited.	[33]
SFN	Non-small-cell lung cancer	In vitro	SREBP1, FASN, ACACA, and ACLY were down-regulated, and the amount of intracellular fatty acids were reduced, leading to apoptosis of human non-small cell lung cancer cells.	[34]
SFN	Pancreatic cancer	In vitro	SFN probably downregulated the sonic hedgehog signaling pathway to prevent pancreatic cancer.	[36]
SFN	Pancreatic cancer	In vitro and in vivo	The stemness of pancreatic CSCs were inhibited by SFN.	[37]
SFN	Epidermal squamous cell carcinoma cancer	In vitro	SFN inhibited ECS formation by reducing levels of YAP1 and ∆Np63α.	[38]
SFN	Breast cancer	In vitro	The expression of exosome miR-140 was increased, and miR-21 and miR-29 were decreased, leading to a decrease in ALDH1 levels and mammosphere formation.	[41]
SFN	Breast cancer	In vitro	The tumorigenicity of MCF10DCIS stem-like cells was inhibited by SFN.	[42]
SFN	Oral squamous cell carcinoma	In vitro and in vivo	SFN intervention resulted in a dose-dependent increase in the levels of tumor suppressive miR-200c.	[43]
SFN	Glioblastoma	In vitro	SFN suppressed PD-L1 expression in a dose-dependent manner.	[47]
SFN	Endometrial cancer	In vitro	SFN inhibited AKT, mTOR, and induced downstream signaling changes in ERK to reduce cancer cell activity.	[50]
SFN	Bladder cancer, hepatoma and colon cancer	In vitro	SFN (1–5 µM) promoted cell growth by 20–43% compared to the control, whereas SFN (10–40 µM) inhibited cell growth in numerous tumor cell models.	[58]
SFN	Liver cancer	In vitro	SFN played an anti-hepatocellular carcinoma role by inhibiting the STAT3/HIF-1alpha/VEGF pathway.	[59]
AITC	Hepatoma	In vitro	AITC (2.5 µM) decreased the DNA damage from 21.57 to 12.30%, while AITC (10 and 20 µM) increased it to 36.12% and 47.48%. DNA repair protein Ku70 was involved in this biphasic effect of AITC on DNA integrity.	[59]
SFN combined with selenium	Hepatoma	In vitro	SFN and selenium have a synergistic effect on the upregulation of thioredoxin reductase1.	[60]
AITC	Hepatoma	In vitro	AITC (2.5 µM) promoted cell migration and DNA damage, which were depended on Nrf2 or glutathione synthesis.	[61]
SFN	Angiogenesis	In vitro and in vivo	Low doses of SFN (2.5–10 μM) alone increased angiogenesis and high concentrations of SFN (20–40 μM) inhibited angiogenesis.	[65]

## Data Availability

Not applicable.

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
