# Peer review of "Dietary Isothiocyanates: Novel Insights into the Potential for Cancer Prevention and Therapy"

_ijms, 2023, doi:10.3390/ijms24031962_

Round 1
Reviewer 1 Report
In abstract authors written in abstract "The application of nano-formulations and delivery of isothiocyanates are also discussed in this review."
but in manuscript they have written only one page .
Author Response
Comment 1: In abstract authors written in abstract “The application of nano-formulations and delivery of isothiocyanates are also discussed in this review. But in manuscript they have written only one page.
Response: Thank you very much for your suggestion, which is essential for our manuscript. We have supplemented a short description on page 9, and added a schematic diagram of major ITCs nano-delivery systems for cancer treatment, which was shown in Figure 3.

Reviewer 2 Report
The manuscript (ijms-2103303) is a review paper concerning novel mechanisms of chemopreventive and therapeutic activity of natural isothiocyanates. It is based on extensive database searching. It is interesting, well-written and valuable. In my opinion the manuscript is worth to be published in IJMS after revisions.
Specific comments:
- More data concerning natural sources of isothiocyanates and examples of the most common isothiocyanates are necessery.
- Figure 1 as a short presentation of novel preventive mechanisms of dietary isothiocyanates should be place at the begining of the text (e.g. just after the Introduction).
- The abbreviation BBN is explained far from the first appearance in the text (pages 3 and 5).
- List of references is partly not prepared according to the journal requirements:
- unnecessary capital letters in all words of the titles of some article - see e.g. ref. # 73, 74, 75
- whole name of the journal name in some references – see e.g. ref. #66, 67
- uncorrect names of the authors of some articles e.g. in ref. # 68 (they should be: Krug P, Mielczarek L, Wiktorska K, Kaczyńska K, Wojciechowski P, Andrzejewski K, Ofiara K, Szterk A, Mazur M) and in ref. #17 (they should be: Angelino D, Jeffery E)
Author Response
Responses to reviewer #2: The manuscript (ijms-2103303) is a review paper concerning novel mechanisms of chemo-preventive and therapeutic activity of natural isothiocyanates. It is based on extensive database searching. It is interesting, well-written and valuable. In my opinion the manuscript is worth to be published in IJMS after revisions.
Comment 1: More data concerning natural sources of isothiocyanates and examples of the most common isothiocyanates are necessary.
Response: Thank you very much for your meaningful comment. We have supplemented a short description on page 2, and added a schematic diagram of the dietary source and basic structure of ITCs in Figure 2.
Comment 2: Figure 1 as a short presentation of novel preventive mechanisms of dietary isothiocyanates should be place at the beginning of the text (e.g. just after the Introduction).
Response: Thank you very much for your suggestion. We have moved Figure 1 just after the Introduction. Thanks again for your comment.
Comment 3: The abbreviation BBN is explained far from the first appearance in the text (pages 3 and 5).
Response: Thank you very much for your comment. We are deeply ashamed of this mistake. We have moved Figure 1 just after the Introduction. Thanks again for your comment.
Comment 4: List of references is partly not prepared according to the journal requirements:
- unnecessary capital letters in all words of the titles of some article - see e.g. ref. # 73, 74, 75
- whole name of the journal name in some references – see e.g. ref. #66, 67
- uncorrect names of the authors of some articles e.g. in ref. # 68 (they should be: Krug P, Mielczarek L, Wiktorska K, Kaczyńska K, Wojciechowski P, Andrzejewski K, Ofiara K, Szterk A, Mazur M) and in ref. #17 (they should be: Angelino D, Jeffery E)
Response: Thank you very much for your comment. We checked all references and marked the corrections in red.
Reviewer 3 Report
The article "Dietary Isothiocyanates: Novel Insights into the Potential for Cancer Prevention and Therapy" presents an interesting study that requires some improvement:
- Include a subsection that describes the fundamental chemistry of isothiocyanates and what aspects make them possess bioactive potential (with a simple figure of the molecule).
- Include a summary table that summarizes the different types of studies (in vitro and in vivo) that demonstrate the bioactive effect described.
Author Response
Responses to reviewer #3: The article "Dietary Isothiocyanates: Novel Insights into the Potential for Cancer Prevention and Therapy" presents an interesting study that requires some improvement:
Comment 1: Include a subsection that describes the fundamental chemistry of isothiocyanates and what aspects make them possess bioactive potential (with a simple figure of the molecule).
Response: Thank you very much for your comment. We have supplemented a short description on page 2, and added a schematic diagram of the dietary source and basic structure of ITCs in Figure 2.
Comment 2: Include a summary table that summarizes the different types of studies (in vitro and in vivo) that demonstrate the bioactive effect described.
Response: Thank you very much for your suggestion. We have supplemented a table (Table 1), which summarized major different types of researches.
Reviewer 4 Report
This is very interesting review article mainly focuses on plant chemical i.e. isothiocyanates which has been considered as healthy died and also play an important role in decreasing the risk of cancer. Authors have added several important findings in this area of research along with well defined concepts of dietary isothiocyanates in several headings. Paper is well written even though, I have some suggestions for the improvement of the text as follows-
1. In the abstract authors have mentioned that Isothiocyanates possess potent anti-cancer activities based on upto-date evidence from in vitro and in vivo studies. Based on this, I will recommend to include a table focusing on in vitro and in vivo studies on dietary isothiocyanates along with dose/ exposure levels. Also, consider possible combinations of dietary isothiocyanates with known anti-cancer drugs.
2. This is very interesting review, however, it would be useful to add a figure focusing on the mechanism of ITCs and how it may effecting cancer. Also, elaborate a role of nanotechnology-based ITC-delivery systems into graphical representation.
3. Author may add Future perspectives along with conclusion.
Author Response
Responses to reviewer #4: This is very interesting review article mainly focuses on plant chemical i.e. isothiocyanates which has been considered as healthy died and also play an important role in decreasing the risk of cancer. Authors have added several important findings in this area of research along with well defined concepts of dietary isothiocyanates in several headings. Paper is well written even though, I have some suggestions for the improvement of the text as follows.
Comment 1: In the abstract authors have mentioned that Isothiocyanates possess potent anti-cancer activities based on upto-date evidence from in vitro and in vivo studies. Based on this, I will recommend to include a table focusing on in vitro and in vivo studies on dietary isothiocyanates along with dose/ exposure levels. Also, consider possible combinations of dietary isothiocyanates with known anti-cancer drugs.
Response: Thank you very much for your suggestion. We have supplemented a table (Table 1), which summarized major different types of researches.
Comment 2: This is very interesting review, however, it would be useful to add a figure focusing on the mechanism of ITCs and how it may effecting cancer. Also, elaborate a role of nanotechnology-based ITC-delivery systems into graphical representation.
Response: Thank you very much for your suggestion. We have supplemented a schematic diagram of major ITCs nano-delivery systems for cancer treatment (Figure 3). And the mechanism of ITCs effecting cancer can be shown in Figure 2.
Comment 3: Author may add Future perspectives along with conclusion.
Response: Thank you very much for your suggestion. Many future perspectives were placed on special part, not along with conclusion. For example, the future perspective of nano-delivery of SFN were placed on section 8 on page 10.
Round 2
Reviewer 1 Report
Kindly cite following reference in text
Yeger H, Mokhtari RB. Perspective on dietary isothiocyanates in the prevention, development and treatment of cancer. J Cancer Metastasis Treat 2020;6:26. http://dx.doi.org/10.20517/2394-4722.2020.61
Author Response
Responses to reviewer #1:
Comment 1: Kindly cite following reference in text.
Response: Thank you very much for your meaningful comment. We have inserted “Yeger H, Mokhtari RB. Perspective on dietary isothiocyanates in the prevention, development and treatment of cancer. J Cancer Metastasis Treat 2020;6:26” as reference No. 5, which adjustment was more appropriate.
Reviewer 3 Report
Accept in present form
Author Response
Responses to reviewer #3:
Comment 1: Accept in present form.
Response: Thank you for your approval of your manuscript, and thank you again for your advice